# Exploring determinants of early marriage among women in Bangladesh: A multilevel analysis

**Md. Mamunur Rashid**[1], **Md. Nure Alam Siddiqi**[1], **Md. Al-Amin**[2], **Md. Mostafizur Rahman**[3], **Tapan Kumar Roy**[3], **Mosiur Rahman**[3], **Md. Jahirul Islam**[4], **Md. Obaidur Rahman**[5,6]*

1 Department of Population Science, Jatiya Kabi Kazi Nazrul Islam University, Mymensingh, Bangladesh, 2 Department of Human Resource Management, Jatiya Kabi Kazi Nazrul Islam University, Mymensingh, Bangladesh, 3 Department of Population Science and Human Resource Development, University of Rajshahi, Rajshahi, Bangladesh, 4 Griffith Criminology Institute, Griffith University, Mount Gravatt, QLD, Australia, 5 Center for Evidence-Based Medicine and Clinical Research, Dhaka, Bangladesh, 6 Department of Global Health Nursing, Graduate School of Nursing Science, St. Luke's International University, Tokyo, Japan

* obaidur006@gmail.com

**Data Availability Statement:** The dataset is publicly available in the Demographic and Health Survey assortment (https://dhsprogram.com/data/available-datasets.cfm).

## Abstract

### Introduction

Early marriage, defined as marriage under the age of 18, is widely recognized as a human rights violation with deleterious consequences on women's health and well-being. It persists as a significant global public health issue, predominantly being practiced in South Asian countries. In Countries like Bangladesh, this practice contributes to an increase in early pregnancies among women of reproductive age, further exacerbating adverse maternal and child health outcomes. While certain predictors of early marriage are recognized, additional investigation is warranted due to diverse socio-economic and demographic circumstances. This study, therefore, aimed to identify the prevalence and determinants of early marriage among women in Bangladesh.

### Methods

This study included a total weighted sample of 18,228 married women aged 18 to 49 years, extracted from the most recent nationally representative Bangladesh Demography and Health Survey (2017–18). We estimated the survey weighted pooled prevalence of early marriage among women and stratified by their different characteristics. We used multilevel mixed-effect binary logistic regression model and estimated the odds ratios (ORs) with their 95% confidence intervals (CIs) to identify the individual-, household-, and community-level factors associated with early marriage practice. All analyses were performed by Stata software version 18.

### Results

Overall, 74.27% [95% CI: 73.15, 75.35] women got married before reaching the age of 18 years. Early marriage was more prevalent in Rajshahi (82.69%), Rangpur (81.35%), and

**Funding:** The author(s) received no specific funding for this work.

**Competing interests:** The authors have declared that no competing interests exist.

Khulna division (79.32%). Women with higher education (OR = 0.10, 95% CI: 0.08, 0.12), husband's higher education (OR = 0.57, 95% CI: 0.48, 0.67), and non-Muslim women (OR = 0.46, 95% CI: 0.40, 0.52) were associated with a lower likelihood of experiencing early marriage. Compared to those household heads aged ≤35 years, the likelihood of early marriage among women was lower for those household heads aged 36–55 years (OR = 0.84, 95% CI: 0.76, 0.93) and >55 years (OR = 0.78, 95% CI: 0.69–0.88). Women aged 18–24 years (OR = 1.24, 95% CI: 1.10, 1.40), husbands with agricultural occupation (OR = 1.22, 95% CI:1.06, 1.41), middle wealth index level (OR = 1.14, 95% CI: 1.02, 1.28), family size five or more (OR = 1.21, 95% CI: 1.11, 1.31), and rural residence (OR = 1.17, 95% CI: 1.04, 1.30) were more likely to experience early marriage.

## Conclusion

This study underscores the alarming prevalence of early marriage among women in Bangladesh, with three-fourths experiencing early marriage, particularly in specific regions. Notably, women education and older household heads were significantly associated with a reduced likelihood of early marriage. Our findings suggest that culturally sensitive interventions should focus on empowering older household heads, alongside initiatives to increase awareness among younger household heads, and enhance education, particularly in rural and impoverished households. These efforts could potentially alter socio-cultural practices and reduce early marriage in Bangladesh.

## Introduction

Early marriage, defined as marriage under the age of 18, is widely acknowledged as a violation of human rights, significantly impacting the health and well-being of women. This practice remains pressing global public health concern, primarily prevalent in South Asian countries. In nations like Bangladesh, it contributes to a surge in early pregnancies and detrimental health outcomes for both mothers and their offspring [1]. The scope of the issue is vast, impacting over 640 million girls and women globally, with elevated occurrences in South Asia (45%), sub-Saharan Africa (20%), and East Asia and the Pacific (15%) [2]. This trend is especially pronounced in developing countries, where one in three females in impoverished nations are being experienced early marriage [2]. Despite significant progress in South Asia, reductions in early marriage are not fast enough to achieve the sustainable development goals (SDG) target of eliminating the practice by 2030, requiring progress to be seven times faster in South Asian countries compared to the last decade [3].

Bangladesh is among the top ten countries for early marriage, with 42 million girls married before the age of 18, accounting for around 7% of the global total [3]. In countries like Bangladesh, the consequences of early marriage are manifold, including gender inequality, adverse reproductive health outcomes, increased instances of intimate partner violence, and contributing to larger family sizes and higher population density [2]. Additionally, it leads to lower educational attainment, undermines women's empowerment, and limits access to reproductive health services, thereby impeding decision-making within households [4]. The lack of information among young girls about pregnancy risks and sexually transmitted diseases perpetuates the cycle of poverty.

Many factors contribute to increase in early marriage practices. A recent systematic review explored several individual-, household-, and community-level factors associated with early marriage among women in low- and middle-income countries (LMICs), including Bangladesh [5]. These factors include the education and occupation of women, their parents, and husbands; household economic status; family size and type; parental decision-making; place of residence; region; ethnicity; and religion. However, most studies in the systematic review are based on small or regional samples that may not be representative, or have focused on specific age ranges, such as adolescents or younger women, potentially leading to selection bias. For example- a recent study explored the geographical variations in early marriage and its predictors in Bangladesh, but only studied women aged 20–24 years from the Bangladesh Demographic and Health Survey (BDHS) 2017–18 data [6]. Furthermore, methodological issues persist in earlier research. Most previous studies have examined the determinants of early marriage using logistic regression models, which may limit the consideration of regional variation when data are clustered. Women within the same cluster may be more similar to each other than those in other clusters, necessitating a multilevel model to estimate comprehensive determinants affecting early marriage and measure cluster variance. This approach would enable a more accurate understanding of the factors influencing early marriage and the extent to which these factors vary across different regions.

While several individual-, household-, and community-level factors have been studied in relation to early marriage in Bangladesh and other LMICs [5–10], additional factors like household headship and the age of the household head need further examination as these factors may influence decision-making in early marriage. To the best of our knowledge, no research has yet examined how household headship and the age of the household head affect the likelihood of early marriage in Bangladesh and other LMICs, using the nationally representative Demographic and Health Survey (DHS) data. Given the hierarchical nature of the DHS data and considering the additional factors (age and sex of household head), this study employs a multilevel model to identify determinants of early marriage, facilitating an understanding across different levels, and addressing standard errors in multiple logistic regression. Therefore, the specific objectives of this study are to determine: 1) the prevalence of early marriage among women aged 18 to 49 years; and 2) the individual-, household-, and community-level factors associated with early marriage in Bangladesh using the most recent BDHS data (2017–2018). By comprehensively analyzing these factors, this study aims to provide actionable insights to inform decision-making and reduce the occurrence of early marriage and its associated adverse outcomes.

## Materials and methods

### Survey setting

We utilized the most recent nationally representative BDHS 2017–18 data. The National Institute of Population Research and Training (NIPORT), the Medical Education and Family Welfare Division, and the Ministry of Health and Family Welfare collaborated to conduct the survey. The DHS survey was approved by the ICF Institutional Review Board (IRB) and informed consent was obtained from the participants in the survey (https://dhsprogram.com/Methodology/Protecting-the-Privacy-of-DHS-Survey-Respondents.cfm).

The survey employed a robust two-stage stratified sampling approach. Initially, 675 enumeration areas were selected, consisting of 250 urban and 425 rural clusters, using the probability proportional to enumeration area size. Bangladesh Bureau of Statistics (BBS) provided a complete list of enumeration areas based on the 2011 population census. Subsequently, household listing activities were conducted in the selected enumeration areas. Following this, 672

enumeration areas were taken into consideration for data collection after excluding three regions- one urban areas and two rural areas- due to flooding concerns. In the subsequent stage, 30 households were selected from each cluster using systematic sampling approach. However, 19,457 households participated in the interview process, with a 99.4% response rate. Among the selected households, 20,376 women were deemed eligible for the survey. Finally, a total of 20,127 ever-married women were interviewed in this survey. The detailed methodology of the survey has been published elsewhere [11].

### Analytical sample

A total weighted sample of 18,228 ever-married women aged 18 to 49 years at the time of the survey was included in this study. We excluded women under the age of 18 from our analysis because their inclusion could skew the data, leading to biased estimates of early marriage rates. Additionally, we found few missing data for the selected explanatory variables (husband's education and occupation) and excluded those cases as well, thereby enhancing the robustness of the analysis.

### Outcome variable

We considered the occurrence of early marriage among women as the outcome variable, defined as a dichotomous variable (1 = yes, 0 = no). In the BDHS 2017–18, women were asked "How old were you when you first started living with your husband?". The outcome variable was derived from this question and classified as early marriage (1 = yes) if the women cohabitated with their husband before reaching the age of 18 years; otherwise, it was classified as not early marriage (0 = no).

### Explanatory variables

In this study, we considered several individual-, household-, and community-level factors, those were selected based on 1) a rapid review of similar published articles or systematic review with or without meta-analyses, including experts' experience and their relevance to the outcome variable; and 2) their availability in the BDHS 2017–18 dataset. We first conducted a rapid review on early marriage by performing a comprehensive search of several electronic databases including PubMed, Medline, Embase, and CINAHL, and identified the relevant studies conducted in Bangladesh and other LMICs [5–10]. We then sorted out individual-, household-, and community-level factors associated with early marriage, that were available in the BDHS 2017–18 dataset.

In our analysis, we considered the following individual-level factors: women's age (18–24, 25–34, and 35–49 years), women's education (no education, primary, secondary, higher), husband's education (no education, primary, secondary, higher), and husband's occupation (services, agriculture, manual labor, business, others). Household-level factors included wealth index (poor, middle, rich), sex of the household head (male, female), age of household head (≤35, 36–55, >55 years), and family size (4 or less, 5 or more). Community-level factors included religion (Muslim, non-Muslim), place of residence (urban, rural), and division (Barisal, Chittagong, Dhaka, Khulna, Mymensingh, Rajshahi, Rangpur, Sylhet).

### Statistical analysis

We weighted the survey data at the beginning of our analysis to ensure the representative sample and to adjust for the non-proportional allocation of the sample to different clusters, thereby providing precise estimates and standard errors. Descriptive statistics were used to describe

the characteristics of the study population. We estimated the survey-weighted pooled prevalence of early marriage among women and stratified by their different characteristics. To visualize the spatial variations, we presented the pooled prevalence of early marriage in eight administrative divisions and produced a division-wise map for early marriage in Bangladesh. We extracted the geographical coordinates in the shapefiles from the DIVA-GIS database (https://diva-gis.org/), and obtained the latitude and longitude for each division in Bangladesh, then produced the map using Stata software.

Due to hierarchical nature of the BDHS data (women/respondents nested within households, and households nested within clusters), we applied a multilevel mixed-effect binary logistic regression model to identify the individual-, household-, and community-level factors associated with early marriage, and estimated the odds ratios (ORs) with their 95% confidence intervals (CIs). Within a cluster of women, there may be similarities in their characteristics, leading to non-independent observations and unequal variance between clusters. In such cases, a multilevel model is very useful as it accounts for the differences between clusters, proving more precise estimates. In our model, we assumed that each community has a unique intercept and fixed coefficients, with random effects applied at the cluster level.

In this study, we fitted five models: the null model with no explanatory variables (Model 0), Model 1 with individual-level factors, Model 2 with household-level factors, Model 3 with community-level factors, and Model 4 with all individual-, household-, and community-level factors. Furthermore, we estimated intra-class correlation coefficient (ICC), Akaike Information Criteria (AIC), and Bayesian Information Criteria (BIC) for model comparison and to measure the goodness of fit. The model with lower AIC and BIC values was considered a better fit for the data. Using variance inflation factor, we also checked the multicollinearity among the explanatory variables before fitting the models and found no concerns with multicollinearity. All analyses were performed by Stata software version 18.0 MP (StataCorp LLC, College Station, Texas, USA).

## Ethical considerations

We used a secondary dataset from the DHS program, which was available to the public domain (https://dhsprogram.com/). Therefore, ethical approval was not required for this study. An explanation of the ethical procedures is also available in the BDHS reports [11].

## Results

A total weighted sample of 18,228 ever-married women aged 18 to 49 years was included in this study (Table 1). Among them, 16.05% women had no formal education. Most of the women's husbands were involved in manual labor (34.07%), agriculture (28.11%), or business (18.15%). More than half of the women belonged to the poor (37.78%) or middle wealth index (20.20%), with 71.41% residing in rural areas. The majority of household heads were male (87.68%) and followed Muslim religion (90.43%). Approximately one-third of the household heads were aged 35 years or younger (29.67%).

Overall, 74.27% [95% CI: 73.15, 75.35] women got married before reaching the age of 18 years. Early marriage was more prevalent in Rajshahi (82.69%), Rangpur (81.35%), and Khulna division (79.32%) (Fig 1), as well as among those residing in Rural areas (77.09%). The prevalence of early marriage was higher for women with a poor wealth index (81.38%), Muslim women (76.07%), and in households where the head was aged 35 years or younger (76.18%). Conversely, a lower prevalence of early marriage was observed among women with higher education (34.14%), husbands with higher education (49.22%), those with a rich wealth index

**Table 1. Prevalence of early marriage with their 95% confidence intervals, stratified by individual-, household-, and community-level characteristics, BDHS 2017–18.**

| | Weighted sample (%) | Prevalence | 95% CIs | |
|---|---|---|---|---|
| | | | Lower | Upper |
| **N** | **18,228** | | | |
| **Early marriage (overall)** | 13,537 | 74.27 | 73.15 | 75.35 |
| **Women's age** | | | | |
| 18–24 years | 4,739 (26.00%) | 73.01 | 71.31 | 74.63 |
| 25–34 years | 6,723 (36.88%) | 71.19 | 69.58 | 72.75 |
| 35–49 years | 6,766 (37.12%) | 78.21 | 76.68 | 79.66 |
| **Women's education** | | | | |
| No education | 2,926 (16.05%) | 86.52 | 84.68 | 88.18 |
| Primary | 5,741 (31.50%) | 82.29 | 80.86 | 83.64 |
| Secondary | 7,160 (39.28%) | 76.28 | 74.86 | 77.63 |
| Higher | 2,401 (13.17%) | 34.14 | 32.00 | 36.34 |
| **Husband's education** | | | | |
| No education | 4,032 (22.12%) | 85.50 | 84.00 | 86.88 |
| Primary | 5,802 (31.83%) | 80.43 | 78.93 | 81.85 |
| Secondary | 5,411 (29.68%) | 73.09 | 71.47 | 74.64 |
| Higher | 2,982 (16.36%) | 49.22 | 46.91 | 51.54 |
| **Husband's occupation** | | | | |
| Services | 2,102 (11.53%) | 73.08 | 70.74 | 75.30 |
| Agriculture | 5,123 (28.11%) | 81.81 | 79.71 | 83.73 |
| Manual labor | 6,210 (34.07%) | 75.33 | 73.77 | 76.83 |
| Business | 3,308 (18.15%) | 72.67 | 70.82 | 74.46 |
| Others | 1,486 (8.15%) | 49.04 | 46.05 | 52.04 |
| **Wealth index** | | | | |
| Poor | 6,886 (37.78%) | 81.38 | 79.60 | 83.03 |
| Middle | 3,681 (20.20%) | 78.70 | 77.05 | 80.27 |
| Rich | 7,661 (42.03%) | 65.75 | 64.09 | 67.36 |
| **Sex of HH** | | | | |
| Male | 15,981 (87.68%) | 74.22 | 73.07 | 75.33 |
| Female | 2,246 (12.32%) | 74.63 | 72.20 | 76.92 |
| **Age of HH** | | | | |
| < = 35 years | 5,407 (29.67%) | 76.18 | 74.62 | 77.67 |
| 36–55 years | 8,961 (49.16%) | 74.77 | 73.38 | 76.10 |
| >55 years | 3,859 (21.17%) | 70.43 | 68.56 | 72.24 |
| **Family size** | | | | |
| 4 or less | 8,096 (44.42%) | 74.12 | 72.67 | 75.52 |
| 5 or more | 10,132 (55.58%) | 74.39 | 73.08 | 75.65 |
| **Religion** | | | | |
| Muslim | 16,483 (90.43%) | 76.07 | 75.05 | 77.06 |
| Non-Muslim | 1,745 (9.57%) | 57.25 | 52.64 | 61.74 |
| **Place of residence** | | | | |
| Urban | 5,212 (28.59%) | 67.23 | 65.09 | 69.30 |
| Rural | 13,016 (71.41%) | 77.09 | 75.77 | 78.35 |
| **Division** | | | | |
| Barisal | 1,015 (5.57%) | 78.44 | 75.81 | 80.85 |
| Chittagong | 3,272 (17.95%) | 69.61 | 66.57 | 72.50 |

*(Continued)*

**Table 1.** (Continued)

| | Weighted sample (%) | Prevalence | 95% CIs | |
|---|---|---|---|---|
| | | | Lower | Upper |
| N | 18,228 | | | |
| Dhaka | 4,688 (25.72%) | 70.10 | 67.11 | 72.93 |
| Khulna | 2,125 (11.66%) | 79.32 | 76.57 | 81.82 |
| Mymensingh | 1,414 (7.76%) | 77.84 | 75.56 | 79.96 |
| Rajshahi | 2,513 (13.78%) | 82.69 | 80.68 | 84.53 |
| Rangpur | 2,146 (11.77%) | 81.35 | 78.85 | 83.62 |
| Sylhet | 1,054 (5.78%) | 53.74 | 49.38 | 58.05 |

Note:

BDHS: Bangladesh Demographic and Health Survey.

CI: Confidence intervals.

HH: Household head.

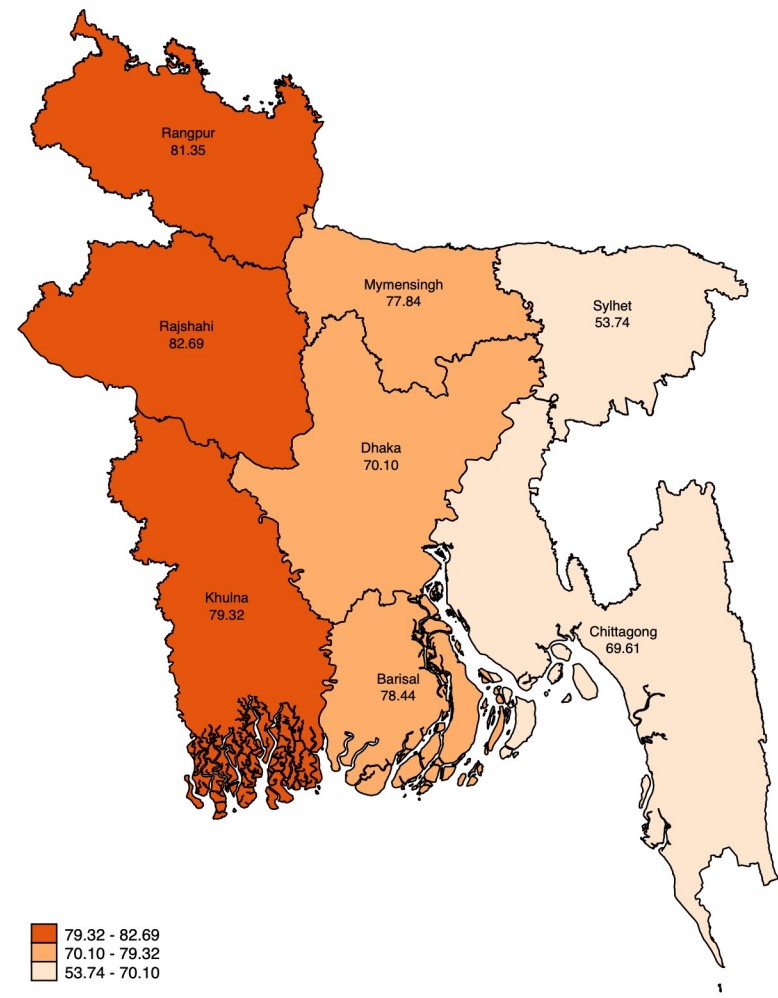

**Fig 1. Prevalence of early marriage in eight administrative regions of Bangladesh.**

(65.75%), and non-Muslim women (57.25%). No significant difference in occurrence of early marriage was observed based on the sex of the household head or family size (Table 1).

Table 2 presents the association of early marriage with individual-, household-, and community-level factors using multilevel mixed-effect binary logistic regression models. The cluster-level variance in the null model (Model 0) was 0.59 (95% CI: 0.50–0.69), and the ICC values was 15.16%, indicating that variations in early marriage among women were largely attributable to differences at the cluster level, while the remaining 84.84% were attributed to individual factors. The ICC value gradually decreased from 15.16% in the null model to 4.59% in the final model (Model 4) after adjusting all individual-, household-, and community-level factors. Furthermore, the AIC and BIC values in the Model 4 were the lowest among all the models, indicating that this model provided the best fit for the explanatory variables predicting early marriage in Bangladesh.

Our best fitted model (Model 4) found that women aged 18–24 years had a 1.24 times higher likelihood (OR = 1.14, 95% CI: 1.10, 1.40) of experiencing early marriage than women aged 35–49 years. There was an inverse association between women/husband education and the likelihood of experiencing early marriage. Women with primary (OR = 0.74, 95% CI: 0.65, 0.86), secondary (OR = 0.56, 95% CI: 0.48, 0.65) and higher education (OR = 0.10, 95% CI: 0.08, 0.12) had a lower likelihood of experiencing early marriage, compared to women with no formal education. Similarly, compared to husbands with no formal education, the likelihood of early marriage was lower for husbands with primary (OR = 0.83, 95% CI: 0.74, 0.94), secondary (OR = 0.74, 95% CI: 0.65, 0.84), and higher education (OR = 0.57, 95% CI: 0.48, 0.67). Moreover, a higher likelihood of early marriage (OR: 1.22, 95% CI: 1.06, 1.41) were found for women whose husbands were involved in agricultural occupation, compared to those whose husbands were involved in service.

Women with a middle wealth index level had 1.14 times higher likelihood of experiencing early marriage (OR = 1.14, 95% CI: 1.02, 1.28) than women with a rich wealth index level. Compared to those household heads aged ≤35 years, the likelihood of early marriage among women was lower for those household heads aged 36–55 years (OR = 0.84, 95% CI: 0.76, 0.93) and >55 years (OR = 0.78, 95% CI: 0.69–0.88). The likelihood of early marriage was 1.21 times higher (OR = 1.21, 95% CI: 1.11, 1.31) among women living in large families (family size 5 or more) than women in small families (family size 4 or less).

Non-Muslim women were associated with a lower likelihood of experiencing early marriage than Muslim women (OR = 0.46, 95% CI: 0.40, 0.52). However, women living in rural areas were 1.17 times more likely to experience early marriage compared to those in urban areas (OR = 1.17, 95% CI: 1.04, 1.30). Furthermore, women living in Rajshahi division (OR = 1.27, 95% CI: 1.03, 1.57) and Rangpur division (OR = 1.29, 95% CI: 1.05, 1.60) had a significantly higher likelihood of early marriage than those living in Barisal division.

## Discussion

In this study, we comprehensively assessed early marriage practice among women aged 18 to 49 years and their individual-, household-, and community-level factors associated with early marriage in Bangladesh using the most recent BDHS data (2017–2018). Our study underscores the alarmingly high prevalence of early marriage among women in Bangladesh, revealing that 74.27% of women were married before reaching the age of 18, consistent with prior studies conducted in Bangladesh [12,13]. This prevalence is particularly pronounced in the Rajshahi, Rangpur, and Khulna division, indicating significant regional disparities in Bangladesh. A recent study, based on the BDHS 2017–18 data, reported similar geographical variations in early marriage among women aged 20–24 years in Bangladesh [6]. These regional variations

**Table 2. Odds ratios with their 95% confidence intervals of early marriage with individual-, household-, and community-level factors among married women in Bangladesh: Multilevel mixed-effect logistic regression analysis.**

| Variables | Model 0 | Model 1 | Model 2 | Model 3 | Model 4 |
|---|---|---|---|---|---|
| **Women's age** | | | | | |
| 18–24 years | | 1.32 [1.19,1.48] ** | | | 1.24 [1.10, 1.40] ** |
| 25–34 years | | 0.94 [0.86, 1.03] | | | 0.89 [0.80,0.98] * |
| 35–49 years [ref] | | | | | |
| **Women's education** | | | | | |
| No education [ref] | | | | | |
| Primary | | 0.76 [0.66, 0.87] ** | | | 0.74 [0.65, 0.86] ** |
| Secondary | | 0.57 [0.49, 0.67] ** | | | 0.56 [0.48,0.65] ** |
| Higher | | 0.10 [0.08,0.12] ** | | | 0.10 [0.08, 0.12] ** |
| **Husband's education** | | | | | |
| No education [ref] | | | | | |
| Primary | | 0.85 [0.75, 0.96] ** | | | 0.83 [0.74, 0.94] ** |
| Secondary | | 0.74 [0.65, 0.84] ** | | | 0.74 [0.65, 0.84] ** |
| Higher | | 0.58 [0.49, 0.68] ** | | | 0.57 [0.48, 0.67] ** |
| **Husband's occupation** | | | | | |
| Services [ref] | | | | | |
| Agriculture | | 1.30 [1.13, 1.50] ** | | | 1.22 [1.06, 1.41] ** |
| Manual labor | | 1.09 [0.96, 1.24] | | | 1.07 [0.94, 1.22] |
| Business | | 1.10 [0.95, 1.26] | | | 1.10 [0.95, 1.26] |
| Others | | 0.93 [0.78, 1.11] | | | 0.94 [0.79, 1.12] |
| **Wealth index** | | | | | |
| Poor | | | 2.38 [2.17, 2.62] ** | | 0.99 [0.88, 1.11] |
| Middle | | | 1.96 [1.76, 2.17] ** | | 1.14 [1.02, 1.28] * |
| Rich [ref] | | | | | |
| **Sex of HH** | | | | | |
| Male [ref] | | | | | |
| Female | | | 1.01 [0.90, 1.13] | | 1.03 [0.92, 1.17] |
| **Age of HH** | | | | | |
| ≤35 years [ref] | | | | | |
| 36–55 years | | | 0.92 [0.84, 1.00] | | 0.84 [0.76, 0.93] ** |
| >55 years | | | 0.79 [0.71, 0.88] ** | | 0.78 [0.69, 0.88] ** |
| **Family size** | | | | | |
| 4 or less [ref] | | | | | |
| 5 or more | | | 1.12 [1.04, 1.21] ** | | 1.21 [1.11, 1.31] ** |
| **Religion** | | | | | |
| Muslim [ref] | | | | | |
| Non-Muslim | | | | 0.46 [0.41, 0.53] ** | 0.46 [0.40, 0.52] ** |
| **Place of residence** | | | | | |
| Urban [ref] | | | | | |
| Rural | | | | 1.73 [1.55, 1.94] ** | 1.17 [1.04, 1.30] ** |
| **Division** | | | | | |
| Barisal [ref] | | | | | |
| Chittagong | | | | 0.69 [0.56, 0.86] ** | 0.60 [0.49, 0.73] ** |
| Dhaka | | | | 0.79 [0.64, 0.98] * | 0.67 [0.55, 0.81] ** |
| Khulna | | | | 1.19 [0.95, 1.49] | 1.18 [0.96, 1.46] |
| Mymensingh | | | | 0.95 [0.76, 1.20] | 0.84 [0.68, 1.03] |

*(Continued)*

**Table 2.** (Continued)

| Variables | Model 0 | Model 1 | Model 2 | Model 3 | Model 4 |
|---|---|---|---|---|---|
| Rajshahi | | | | 1.38 [1.10, 1.72] ** | 1.27 [1.03, 1.57] * |
| Rangpur | | | | 1.31 [1.04, 1.64] * | 1.29 [1.05, 1.60] * |
| Sylhet | | | | 0.35 [0.28, 0.44] ** | 0.24 [0.19, 0.29] ** |
| **Random effects** | | | | | |
| Cluster-level variance | 0.59 [0.50, 0.69] | 0.50 [0.43, 0.59] | 0.44 [0.37, 0.52] | 0.27 [0.22, 0.33] | 0.16 [0.12, 0.21] |
| Intra-class correlation | 15.16% | 13.23% | 11.86% | 7.63% | 4.59% |
| **Goodness of fit** | | | | | |
| AIC | 20372.67 | 18152.22 | 20002.72 | 19999.84 | 17680.32 |
| BIC | 20388.29 | 18261.54 | 20065.19 | 20085.73 | 17906.76 |

Note:

** $p < .01$

* $p < a .05$.

Model 0: Null model.

Model 1: Adjusted for individual-level factors such as women's age and education, husband's education and occupation.

Model 2: Adjusted for household-level factors such as wealth index, sex and age of HH head, and family size.

Model 3: Adjusted for community-level factors such as religion, place of residence, and division.

Model 4: Adjusted for individual-, household-, and community-level factors.

AIC: Akaike's information criterion.

BIC: Bayesian information criterion.

HH: Household head.

Ref: Reference category.

indicate that local socio-cultural norms, community awareness levels, and economic conditions are attributable to influencing early marriage practices [7,14]. Addressing these regional variations requires targeted interventions that consider the socio-cultural contexts of each region.

Our multilevel analysis identified several individual-, household-, and community-level determinants of early marriage among women. Notably, age of the household heads was significant factors; women with older household heads were less likely to experience early marriage, suggesting that older household heads may possess more maturity and awareness regarding the adverse effects of early marriages. In LMICs like Bangladesh, young household heads (aged 35 years and below) are more likely to permit early marriages due to their poverty, lack of education, and immature decision-making. Conversely, older household heads are more informed and influenced by awareness programs from government and non-government organizations, enabling them to make better decisions regarding their children's education and marriage, thus contributing to the reduction of early marriage [15,16]. Therefore, our findings suggest that empowering older household heads and advancing knowledge among younger household heads through awareness-raising programs could be a viable strategy to reduce early marriage practices in resource-limited settings like Bangladesh.

The sex of the household head is an influential factor of early marriage. A recent study observed that the occurrence of early marriage in LMICs was higher among male-headed households, compared to female-headed households [17]. However, our study did not find any significant difference in early marriage practices between male- and female-headed households, possibly due to the limited number of families headed by females in our sample.

Our study identified a strong inverse relationship between educational attainment (both for women and their husbands) and the likelihood of early marriage. Higher educational levels

were associated with a significantly lower likelihood of early marriage, highlighting the critical role of education in delaying marriage. This finding aligns with existing literatures, which emphasize education as a protective factor against early marriage [18–21]. Low educational attainment for women not only deprives them of their fundamental rights but also increases the prevalence of early marriage, leading to adverse reproductive, maternal, and child health outcomes [15]. On the other hand, husbands with higher education are more aware of the negative consequences of early marriage and likelihood to delay marriage [12]. Delaying the age at first marriage through higher education can contribute to professional advancement and societal development, as well as to curb early marriage. Furthermore, husband occupation influenced early marriage, with higher rates among those in agricultural occupations due to lower education levels and poverty [5,22]. Therefore, policies aimed at improving access to education, particularly for girls, are essential in mitigating early marriage practices.

Similar to prior studies, a higher prevalence of early marriage was observed among Muslim families, compared to non-Muslim families, likely due to persistent religious and cultural norms [23,24]. Furthermore, Muslim women from larger families were more likely to experience early marriage, as girls are often seen as economic burdens in financially strained households, leading parents to marry them off at a very young age [9,25]. Therefore, our findings support culturally-appropriate and effective interventions to reduce early marriage practices in Muslim families.

Early marriage was more prevalent among rural women than urban women, which is consistent with prior studies [26,27]. Rural women may lack awareness of the adverse impacts of early marriage, highlighting the need for targeted programs to empower women and raise awareness in these areas. Various factors such as economic conditions, socio-demographics, traditional norms, and cultural beliefs may contribute to these differences. Poverty and illiteracy are key reasons for higher early marriage rates in rural areas, including certain administrative regions like Rangpur and Rajshahi divisions. In Rangpur, poverty and illiteracy rates are higher than the Sylhet division, significantly impacting the frequency of early marriage [28]. Furthermore, the ethnic communities in the Rajshahi division are more prevalent. The timing of the first marriage might be influenced by demographic, economic and socio-cultural variations among different communities in the Rajshahi division. Conversely, the low poverty rate and greater awareness of the negative consequences of early marriage in the Sylhet division contribute to its lower rate [28].

Women's economic status was identified as another determinant of early marriage, with a negative relationship observed between wealth index and age at the first marriage. Early marriage decreased among economically well-off women and increased among economically disadvantaged ones, consistent with findings in other LMICs [17,29]. Furthermore, women from the poor wealth index may encounter barriers to completing higher education and lack adequate knowledge of reproductive health. Consequently, they may struggle to make well-informed decisions regarding their sexual and reproductive health. Economic empowerment initiatives, such as increasing employment opportunities and providing training, could help to reduce early marriage in Bangladesh.

## Strengths and limitations of the study

This study has several strengths. First, we analyzed a large and nationally representative sample of ever-married women aged 18 to 49 from the most recent BDHS data (2017–18), ensuring the generalizability of our findings in the similar settings like Bangladesh and other LMICs. Second, we used a multilevel mixed-effect binary logistic regression model to identify the individual-, household-, and community-level factors associated with early marriage. This robust

methodological approach accounts for the hierarchical structure of the data, providing more precise estimates by considering cluster-level variations. Additionally, along with several important individual-, household-, and community-level factors, this study comprehensively assessed the role of age and sex of the household heads on early marriage, that have not been previously studied using nationally representative DHS data. Survey weighting and stratification by various characteristics enhanced the reliability and accuracy of the prevalence estimates and identified associations. Furthermore, the study highlighted regional variations in the prevalence of early marriage, providing valuable geographical insights that can inform targeted interventions.

This study also has limitations that need to be considered when interpreting our findings. First, the data used in this study are cross-sectional, that limits the ability to infer causality between the identified factors and early marriage, as the associations observed do not establish temporal relationships. Second, the reliance on self-reported data for the age at first marriage may introduce recall or social desirability bias. Furthermore, some influential factors such as cultural practices, decision regarding the first marriage, or detailed parental influence, were not included due to data limitations. Although we weighted the sample before analysis, a few missing data in explanatory variables could influence our estimates. Lastly, the study lacks qualitative data that could provide a deeper understanding of the socio-cultural context, including the role of religious leaders, and personal experience related to early marriage.

## Conclusion

This study highlights a high prevalence of early marriage among women in Bangladesh, with three-fourths experiencing early marriage, particularly in specific regions such as Rajshahi, Rangpur, and Khulna division. The occurrence of early marriage is notably higher in rural areas and among women with lower socio-economic status. Conversely, older household heads, non-Muslim women, and higher educational levels in both women and their husbands were significantly associated with a reduced likelihood of early marriage. Our findings suggest that culturally appropriate and effective interventions should focus on empowering older household heads, increasing awareness among younger household heads, and enhancing education, particularly in rural and impoverished households. These efforts could potentially alter socio-cultural practices and contribute to reducing early marriage among women in Bangladesh. By providing valuable insights into the underlying individual-, household-, and community-level factors of early marriage, our study can inform decision-making and significantly contribute towards achieving the SDG target of eliminating early marriage by 2030.

## Supporting information

**S1 File. STROBE statement-checklist for observational studies.**
(DOCX)

## Acknowledgments

We are grateful to the DHS programs, for the permission to use all the relevant DHS data for this study. We would like to thank St. Luke's International University, Tokyo, Japan for providing access to their library for database searching and acquisition of relevant articles for our rapid review.

## Author Contributions

**Conceptualization:** Md. Mamunur Rashid.

**Data curation:** Md. Mamunur Rashid, Md. Nure Alam Siddiqi, Md. Obaidur Rahman.

**Formal analysis:** Md. Mamunur Rashid, Md. Obaidur Rahman.

**Investigation:** Md. Mamunur Rashid, Md. Nure Alam Siddiqi, Md. Obaidur Rahman.

**Methodology:** Md. Mamunur Rashid, Md. Nure Alam Siddiqi, Md. Obaidur Rahman.

**Project administration:** Md. Obaidur Rahman.

**Software:** Md. Obaidur Rahman.

**Supervision:** Md. Obaidur Rahman.

**Validation:** Md. Obaidur Rahman.

**Visualization:** Md. Mamunur Rashid, Md. Nure Alam Siddiqi, Md. Obaidur Rahman.

**Writing – original draft:** Md. Mamunur Rashid, Md. Nure Alam Siddiqi.

**Writing – review & editing:** Md. Al-Amin, Md. Mostafizur Rahman, Tapan Kumar Roy, Mosiur Rahman, Md. Jahirul Islam, Md. Obaidur Rahman.

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
