## [Decision Letter · Decision Letter 0]

10 Mar 2024

PONE-D-23-15427Exploring Determinants of Early Marriage among Women in Bangladesh: A Multilevel AnalysisPLOS ONE

Dear Dr. Rashid,

Thank you for submitting your manuscript to PLOS ONE. After careful consideration, we feel that it has merit but does not fully meet PLOS ONE’s publication criteria as it currently stands. Therefore, we invite you to submit a revised version of the manuscript that addresses the points raised during the review process.

Given the concerns from reviewers and the editor, I recommend that the manuscript undergo major revisions before being reconsidered for publication in this journal. The authors should address the issues related to novelty, methodological innovation, variable selection, and rationale as outlined in this review. Additionally, the manuscript should clearly differentiate the research from existing studies and highlight its unique contributions to the field. Please note that acceptance of the paper is not guaranteed unless these concerns are adequately addressed.

We look forward to receiving your revised manuscript.

Kind regards,

Enamul Kabir

Academic Editor

PLOS ONE

A clean copy of the edited manuscript (uploaded as the new *manuscript* file)”.

 [The funders had no role in study design, data collection and analysis, decision to publish, or preparation of the manuscript.]. 

Additional Editor Comments:

The manuscript presents an analysis of factors influencing early marriage in Bangladesh. The authors utilize multilevel logistic regression to examine the impact of various socio-demographic factors on the occurrence of early marriage. While the study provides valuable insights into this important social issue, there are several concerns that need to be addressed before it can be considered for publication in this journal.

The novelty of the research is questionable as the identified factors such as age, education, place of residence, division, wealth index, and religion have been previously documented in research conducted in Bangladesh. The authors need to demonstrate how their study contributes to the existing literature by introducing methodological or design innovations that differentiate it from previous research. The use of multilevel logistic regression, referred to as multinomial logistic regression in the manuscript, is not adequately justified and does not contribute significantly to the novelty of the study.

The inclusion of independent variables in the analysis is not clearly explained, and there is a risk of important potential variables being omitted. The authors should consider utilizing association rule mining and other variable selection methods to ensure that all relevant variables are included in the model. Additionally, the reference category in Table 2 should be adjusted to improve the interpretability of the results, specially making ages 35-49 as reference category.

The rationale for the research is not convincingly presented, and the findings do not offer substantial insights beyond what is already known from previous studies. The authors should articulate a clearer rationale for their research and highlight any unique contributions or insights that distinguish their study from existing literature. Differentiating the research by examining trends from the early 20th century and identifying changes in factors over recent surveys could enhance the significance of the study.

Reviewers' comments:

Reviewer's Responses to Questions

**Comments to the Author**

1. Is the manuscript technically sound, and do the data support the conclusions?

Reviewer #1: Partly

Reviewer #2: Yes

2. Has the statistical analysis been performed appropriately and rigorously? 

Reviewer #1: Yes

Reviewer #2: No

3. Have the authors made all data underlying the findings in their manuscript fully available?

Reviewer #1: Yes

Reviewer #2: Yes

4. Is the manuscript presented in an intelligible fashion and written in standard English?

Reviewer #1: Yes

Reviewer #2: No

5. Review Comments to the Author

Reviewer #1: Abstract: Please define the early marriage in the methods section.

Line 11: Since the aim of this study was to identify prevalence and determinants, I would suggest to mention the prevalence of early marriage first then determinants

Line 13: can we say effect in a cross-sectional study? Please use appropriate term

Line 18: high is a vague term. You didn’t mention any percentage. So how the reader will understand high prevalence?

Lines 58-63: The rational of this study need to be strengthened. Is the multilevel modelling only reason for this study? If so then, using multiple logistic regression adjusting the cluster variables such as administrative division, urban and rural and the primary sampling unit will provide almost the similar results. The authors are advised to make strong rational behind this study.

Lines 70-71: Is this correct? Taken representative sample from entire population. Please rewrite.

Line 107: Is it wise to say risk factor for cross-sectional study?

In the statistical analysis section, explain bit more about the multi-level modelling, the number of levels and which level nested with other levels.

Line 139:140: Revise the sentence.

In Figure 1, the values were not clear in the read zones. Please make it clear

Line 148: please rewrite the sentence.

Line 155: What do you mean by extremely young age? Please clarify

Line 156: effect or association?

Lines 156-158: Make consistent in the categories throughout the manuscript. In the method section, the authors mentioned Poorest and Richest but here poor and rich.

Lines 232-233: Conclusion should be based on the main findings.

Reviewer #2: Comments on the paper

General:

Thank you for allowing me to review your manuscript. I admire the effort and dedication evident in conducting this research and compiling the findings. From my point of view, the manuscript needs a revision.

1. The abstract effectively summarizes the study objectives and key findings. However, it could benefit from a clearer statement of the study's significance and implications for addressing the issue of early marriage in Bangladesh.

2. Introduction: The introduction provides a comprehensive background on early marriage but lacks a clear statement of the research gap or objectives. It could be improved by explicitly stating the research objectives to provide readers with a clear understanding of the study's purpose.

3. Methods:

Survey Setting: The description of the survey methodology lacks detail regarding the sampling strategy beyond the two-stage stratified sampling approach. Readers would benefit from more information on how clusters were selected, especially regarding the exclusion of certain regions, which could introduce bias.

Analytical Sample: While the total number of women included in the analysis is provided, there is no discussion of how missing data or non-response were handled, which is crucial for assessing the robustness of the findings and potential biases in the sample.

Outcome Variable: The definition of the outcome variable, "age at first cohabitation," is insufficiently explained. Given the sensitivity of this measure, clarity on how it was defined and operationalized is essential for readers to interpret the results accurately.

Explanatory Variables: The rationale for selecting specific explanatory variables is not explicitly stated. Without justification for including factors such as women's age, education, religion, wealth index, etc., readers may question the relevance and comprehensiveness of the analysis.

There is no discussion on the assumptions underlying multilevel modeling or potential limitations associated with its application in this context. Providing such insights would enhance the methodological rigor and transparency of the study.

4. Results:

While the author's rationale for using multilevel logistic regression is understandable, providing a more comprehensive explanation of how this approach effectively addresses the limitations of previous methods would enhance the robustness of the study. Clarifying the specific advantages of multilevel modeling in capturing regional variation would strengthen the methodological justification and improve reader confidence in the chosen analytical approach.

5. The discussion could be further developed by exploring the underlying mechanisms driving these associations and proposing actionable recommendations for policymakers and stakeholders.

6. While the manuscript acknowledges some limitations of the study, such as recall bias, it does not provide a comprehensive discussion of methodological limitations and their implications for interpreting the results.

7. The conclusion section provides a brief summary of the study findings but lacks specific recommendations for policymakers and stakeholders. Without actionable guidance on how to address the issue of early marriage in Bangladesh based on the research findings, the manuscript falls short of its potential to inform policy and practice effectively.

6. PLOS authors have the option to publish the peer review history of their article (what does this mean?). If published, this will include your full peer review and any attached files.

Reviewer #1: No

Reviewer #2: No

---

## [Author Response · Author response to Decision Letter 0]

11 Jun 2024

Title: “Exploring Determinants of Early Marriage among Women in Bangladesh: A Multilevel Analysis”

Manuscript ID: PONE-D-23-15427R1 

Author Response to Editorial Office Comments: 

1. Thank you for stating the following financial disclosure: [The funders had no role in study design, data collection and analysis, decision to publish, or preparation of the manuscript.]. At this time, please address the following queries: 

Author response: Thank you for your queries. We did not receive any funding for this study. Therefore, we would like to amend the statement as follows: “The authors received no specific funding for this work.” We have also included this information in our cover letter. 

2. We note that Figure 1 in your submission contain [map/satellite] images which may be copyrighted. All PLOS content is published under the Creative Commons Attribution License (CC BY 4.0), which means that the manuscript, images, and Supporting Information files will be freely available online, and any third party is permitted to access, download, copy, distribute, and use these materials in any way, even commercially, with proper attribution. For these reasons, we cannot publish previously copyrighted maps or satellite images created using proprietary data, such as Google software (Google Maps, Street View, and Earth). For more information, see our copyright guidelines: http://journals.plos.org/plosone/s/licenses-and-copyright. 

Author response: We understand the copyright issues associated with the map we initially produced using ArcGIS software. To visualize the regional variations of early marriage prevalence among women in Bangladesh, we have now created a new map using STATA software version 18 MP. More specifically, we presented the pooled prevalence of early marriage in eight administrative divisions and produced a division-wise map for early marriage in Bangladesh. We extracted the geographical coordinates in the shapefiles from the DIVA-GIS database (https://diva-gis.org/), and obtained the latitude and longitude for each division in Bangladesh, then produced the map using Stata software version 18 MP. We ensure that there has no copyright issues with this map, as DIVA-GIS provide free spatial data for the whole world. 

Author Response to the Editor(s)' Comments: 

Author response: Thank you very much for your valuable comments and constructive suggestions. We have revised our manuscript accordingly. Please find our response for each comment in a point-by-point manner below. 

Author response: We have followed the guidelines, formatted the manuscript, and renamed our files according to the PLOS ONE style requirements as outlined in the provided templates. 

2. We suggest you thoroughly copyedit your manuscript for language usage, spelling, and grammar. 

Upon resubmission, please provide the following: The name of the colleague or the details of the professional service that edited your manuscript

A clean copy of the edited manuscript (uploaded as the new *manuscript* file)”. 

Author response: We have thoroughly copyedited our manuscript to ensure proper language usage, spelling, and grammar. Dr. Md. Obaidur Rahman, PhD (https://orcid.org/0000-0002-2219-3013), a content and methodology expert, performed this comprehensive copyedit with the assistance of Md. Jahirul Islam, PhD, a content expert, and Assistant professor Md. Nure Alam Siddiqi, a content expert. Due to their significant contributions in re-analyzing the data and drafting or revising the manuscript, we have also shared authorship with them. 

We have submitted the revised manuscript, including both a cleaned version and a highlighted track-changed version, through the journal’s online submission system. 

[The funders had no role in study design, data collection and analysis, decision to publish, or preparation of the manuscript.]. 

Author response: Thank you for your queries. We did not receive any funding for this study. Therefore, we would like to amend the statement as follows: “The authors received no specific funding for this work.” We have also included this information in our cover letter. 

Author response: We understand the copyright issues associated with the map we initially produced using ArcGIS software. To visualize the regional variations of early marriage prevalence among women in Bangladesh, we have now created a new map using STATA software version 18 MP. More specifically, we presented the pooled prevalence of early marriage in eight administrative divisions and produced a division-wise map for early marriage in Bangladesh. We extracted the geographical coordinates in the shapefiles from the DIVA-GIS database (https://diva-gis.org/), and obtained the latitude and longitude for each division in Bangladesh, then produced the map using Stata software version 18 MP. We ensure that there has no copyright issues with this map, as DIVA-GIS provide free spatial data for the whole world. 

Author response: Thank you for your guidance. We have included captions for our supporting information files at the end of the manuscript and updated the in-text citations to match accordingly. 

Author Response to Additional Editor Comments: 

The manuscript presents an analysis of factors influencing early marriage in Bangladesh. The authors utilize multilevel logistic regression to examine the impact of various socio-demographic factors on the occurrence of early marriage. While the study provides valuable insights into this important social issue, there are several concerns that need to be addressed before it can be considered for publication in this journal. 

Author response: Thank you very much for your appreciation, and the constructive comments and suggestions. We have revised our manuscript accordingly. Please find our response for each comment below. 

1. The novelty of the research is questionable as the identified factors such as age, education, place of residence, division, wealth index, and religion have been previously documented in research conducted in Bangladesh. The authors need to demonstrate how their study contributes to the existing literature by introducing methodological or design innovations that differentiate it from previous research. The use of multilevel logistic regression, referred to as multinomial logistic regression in the manuscript, is not adequately justified and does not contribute significantly to the novelty of the study. 

Author response: Thank you very much for your great observation and feedback. We agree with your points. To ensure the scope of this study, we have conducted a rapid review (during this revision period) on early marriage by performing a comprehensive search of several electronic databases including PubMed, Medline, Embase, and CINAHL, and identified the relevant studies conducted in Bangladesh and other low- and middle-income countries (LMICs). We noticed that several individual-, household-, and community-level factors (such as age, education, place of residence, division, wealth index, and religion) have been studied in relation to early marriage in Bangladesh and other LMICs. However, no research has yet examined how household headship and the age of the household head affect the likelihood of early marriage in Bangladesh and other LMICs, using the nationally representative Demographic and Health Survey (DHS) data. Furthermore, most studies were based on small or regional samples that may not be representative, or have focused on specific age ranges, such as adolescents or younger women, potentially leading to selection bias. Given the hierarchical nature of the DHS data and considering the additional factors (age and sex of household head), this study employs a multilevel mixed-effect binary logistic regression model to identify the individual-, household-, and community-level factors associated with early marriage among women aged 18 to 49 years, using the most recent BDHS data (2017-18). This robust methodological approach accounts for the hierarchical structure of the data, providing more precise estimates by considering cluster-level variations. The novelty of the research has been clearly stated in the introduction section (Line 92-125) and strengths and limitations of the study in the discussion section (Line 340-354). By providing valuable insights into the underlying individual-, household-, and community-level factors of early marriage, our study can provide informed decision-making and significantly contribute towards achieving the SDG goal of eliminating early marriage by 2030, in Bangladesh and other similar LMICs.

2. The inclusion of independent variables in the analysis is not clearly explained, and there is a risk of important potential variables being omitted. The authors should consider utilizing association rule mining and other variable selection methods to ensure that all relevant variables are included in the model. 

Author response: Thank you very much for your great suggestion. We have incorporated your suggestions. In this study, we have considered several individual-, household-, and community-level factors, those were selected based on 1) a rapid review of similar published articles or systematic review with or without meta-analyses, including experts’ experience and their relevance to the outcome variable; and 2) their availability in the BDHS 2017-18 dataset. We first conducted a rapid review on early marriage by performing a comprehensive search of several electronic databases including PubMed, Medline, Embase, and CINAHL, and identified the relevant studies conducted in Bangladesh and other LMIC. We then sorted out individual-, household-, and community-level factors associated with early marriage, that were available in the BDHS 2017-18 dataset. We have fitted five models: the null model with no explanatory variables (Model 0), Model 1 with individual-level factors, Model 2 with household-level factors, Model 3 with community-level factors, and Model 4 with all individual-, household-, and community-level factors. Furthermore, we estimated intra-class correlation coefficient (ICC), Akaike Information Criteria (AIC), and Bayesian Information Criteria (BIC) for model comparison and to measure the goodness of fit. The model with lower AIC and BIC values was considered a better fit for the data (Lines 161-170; 197-206). 

3. Additionally, the reference category in Table 2 should be adjusted to improve the interpretability of the results, specially making ages 35-49 as reference category. 

Author response: We have incorporated your comments into our revised manuscript. We changed the reference category in Table 2 where required; specifically, we considered the age group 35–49 as a reference category and performed the analysis again (Table 2). 

4. The rationale for the research is not convincingly presented, and the findings do not offer substantial insights beyond what is already known from previous studies. The authors should articulate a clearer rationale for their research and highlight any unique contributions or insights that distinguish their study from existing literature. Differentiating the research by examining trends from the early 20th century and identifying changes in factors over recent surveys could enhance the significance of the study. 

Author response: As suggested, we have explained the rationale of our manuscript in details into the introduction section (Line 92-125) and emphasized unique contributions or insights that set our work apart from previous research in our revised manuscript (Line 340-354). 

Author Response to the Comments from Reviewer 1: 

1. Abstract: Please define the early marriage in the methods section. 

Author response: Thank you very much for your suggestion. We have defined “early marriage” at the beginning of the introduction section of the abstract as the term “early marriage” appeared first in the place (Line 25). 

2. Line 11: Since the aim of this study was to identify prevalence and determinants, I would suggest to mention the prevalence of early marriage first then determinants 

Author response: We have incorporated your suggestion and mentioned the prevalence of early marriage first then its determinants (Line 31-33). 

“This study, therefore, aimed to identify the prevalence and determinants of early marriage among women in Bangladesh.” 

3. Line 13: Can we say effect in a cross-sectional study? Please use appropriate term. 

Author response: We sincerely apologize for this typo-mistake. Of course, we cannot say effect in a cross-sectional study. As this is a cross-sectional st

---

## [Decision Letter · Decision Letter 1]

14 Oct 2024

Exploring Determinants of Early Marriage among Women in Bangladesh: A Multilevel Analysis

PONE-D-23-15427R1

Dear Dr. RAHMAN,

We’re pleased to inform you that your manuscript has been judged scientifically suitable for publication and will be formally accepted for publication once it meets all outstanding technical requirements.

Kind regards,

Enamul Kabir

Academic Editor

PLOS ONE

Additional Editor Comments (optional):

Reviewers' comments:

Reviewer's Responses to Questions

**Comments to the Author**

1. If the authors have adequately addressed your comments raised in a previous round of review and you feel that this manuscript is now acceptable for publication, you may indicate that here to bypass the “Comments to the Author” section, enter your conflict of interest statement in the “Confidential to Editor” section, and submit your "Accept" recommendation.

Reviewer #1: All comments have been addressed

Reviewer #2: All comments have been addressed

2. Is the manuscript technically sound, and do the data support the conclusions?

Reviewer #1: Yes

Reviewer #2: Yes

3. Has the statistical analysis been performed appropriately and rigorously? 

Reviewer #1: Yes

Reviewer #2: Yes

4. Have the authors made all data underlying the findings in their manuscript fully available?

Reviewer #1: Yes

Reviewer #2: No

5. Is the manuscript presented in an intelligible fashion and written in standard English?

Reviewer #1: Yes

Reviewer #2: Yes

6. Review Comments to the Author

Reviewer #1: The authors have addressed the comments properly. One last concern, why the author included 18-49 years age group rather than 15-49 years? Need a clear justification for this because BDHS included the age group of 15-49 years.

Reviewer #2: Minor issues:

• While the authors have improved the significance of their study, I suggest briefly elaborating on how addressing early marriage could lead to broader societal benefits, such as improved health outcomes, economic development, and gender equality. This would further clarify the broader implications of their findings.

• Analytical Sample:

Specify the percentage of missing data for clarity (e.g., "we found a small percentage of missing data...").

• In the introduction, it might be helpful for the authors to briefly explain why a multilevel model is particularly suitable for this study. A sentence or two outlining the advantages of this approach in understanding the complexities of early marriage could enhance clarity.

• Survey Setting: Authors have made an effort to include more details about the methodology. It would be helpful to clarify how clusters were selected and why certain regions were excluded to fully address potential biases.

7. PLOS authors have the option to publish the peer review history of their article (what does this mean?). If published, this will include your full peer review and any attached files.

Reviewer #1: **Yes: **Md. Tariqujjaman

Reviewer #2: No

---

## [Editor Report · Acceptance letter]

22 Oct 2024

PONE-D-23-15427R1 

PLOS ONE

Dear Dr. Rahman, 

I'm pleased to inform you that your manuscript has been deemed suitable for publication in PLOS ONE. Congratulations! Your manuscript is now being handed over to our production team.

Kind regards, 

on behalf of

Dr. Enamul Kabir 

Academic Editor

PLOS ONE